# Ecological Risk Assessment, Distribution and Source of Polycyclic Aromatic Hydrocarbons in the Soil of Urban and Suburban Forest Areas of Southern Poland

Stanisław Łyszczarz * , Jarosław Lasota and Ewa Błońska 

Department of Ecology and Silviculture, Faculty of Forestry, University of Agriculture in Krakow, 29 Listopada 46, 31-425 Kraków, Poland; jaroslaw.lasota@urk.edu.pl (J.L.); ewa.blonska@urk.edu.pl (E.B.)
* Correspondence: stanislaw.lyszczarz@urk.edu.pl

**Abstract:** Polycyclic aromatic hydrocarbons (*PAHs*) are widespread environmental pollutants that can pose a risk to people living near contaminated soils. The role of forest ecosystems around urban agglomerations as a draw to urban dwellers has been highlighted by the COVID-19 pandemic. The pandemic led us to focus this study on the soils beneath forested areas around urban agglomerations, with the aim of assessing the sources and ecological risk of *PAHs* in the soils. For the study, a 150 km transect was delineated by six sampling sites, located in urban and commercial forests, which were characterised by the same species composition. Samples were taken from the 0–10 cm depth horizon, after removal of the organic layer. The content of 16 *PAHs* was determined, from which the potential source of contamination, the toxic equivalent quotient (*TEQ*), the potential ecological risk quotient (*RQ*) and the incremental lifetime cancer risk quotient (*ILCR*) were calculated. The mean sums of the *PAHs* ranged from 12.41 to 52.76 $\mu g \cdot kg^{-1}$. Our analysis indicated that the share of pollutants in the soils had resulted from industrial combustion, biomass and coal combustion, and traffic. The *RQ* of the *PAHs* in the soils of the Upper Silesian Industrial Region, or within its impact range, were found to be low to moderate. High *ILCR* ratios for children ($1.9 \times 10^{-4}$) and adults ($8.38 \times 10^{-5}$) were recorded in soils related to a refinery. Our findings confirm that forests around urban areas are vulnerable to pollution. People living in cities should consider spending their leisure time in forest areas at a distance from their homes. Systematic and continuous monitoring of *PAHs* is necessary to ensure that human safety is guaranteed.

**Keywords:** emissions sources; forest soils; organic pollutants; COVID-19 pandemic; risk assessment



## 1. Introduction

The coronavirus pandemic has brought various restrictions on access, movement and social behaviour in populations around the world. During this time, fields, forests and water bodies became strongly associated with places people claimed had wellbeing benefits [1]. The COVID-19 pandemic has strongly impacted society, causing drastic changes in people's routines and daily mobility, and showing public spaces in a new light [2]. In many countries, where the use of green spaces was not forbidden, there was an increase in the use of such areas during the lockdown [3]. The forests most visited during the pandemic were those around urban agglomerations, which are the ones most often exposed to negative impacts from human activities. Contamination of the environment with polycyclic aromatic hydrocarbons (*PAHs*), next to heavy metals, is one of the most serious threats to the proper functioning of ecosystems, with *PAHs* being one of the groups of persistent organic pollutants that can exhibit strong toxic, mutagenic and carcinogenic properties [4,5]. They occur in aspects of the environment, such as the air, water, soil, and living organisms, which is related to their formation through the processes of incomplete combustion of organic substances. The majority of these compounds come from anthropogenic sources such as industrial processes related to the combustion of crude oil and coal, room heating, road

transport and the incineration of municipal and industrial waste [6,7]. *PAHs* demonstrate high durability and stability in various environmental conditions [8]. The complex ring structure of *PAHs* ensures resistance to degradation processes and promotes the durability and stability of organic compounds [9]. Due to their hydrophobic lipophilic properties, *PAHs* are preferentially adsorbed onto organic matter in the dissolved phase, forming suspended particulates that affect aquatic organisms and, ultimately, human health through the food chain [10]. A large accumulation of pollutants in soil negatively affects the condition of ecosystems, which also poses a direct threat to human health. It has therefore become necessary to monitor the state of the environment, and especially PAH accumulation.

Current studies are often focused on the assessment of toxicity and the determination of the ecological and health risks caused by *PAHs*. Guidelines for soil and air quality, and indicators of potential ecological risk, are useful methods that allow the assessment of the risk of pollution by *PAHs*. To evaluate the single and fundamental hazards of *PAHs* in ecological environments, several assessment methods and concentration indicators have been proposed and widely applied [11,12]. Understanding the impact of particular emissions sources on the different parts of the environment is crucial for proper risk assessment and risk management. An important tool for the identification of pollution emission sources may be PAH diagnostic ratios [13]. According to Cachada et al. [14], the identification of contaminant sources is a critical step in risk assessment and management, especially in complex environments, such as urban areas, in which there is not one single source, but several point and diffuse sources. The PAH production processes, associated with their source, shape the PAH emissions profiles [15]. By tracing the individual ratios of PAH compounds, the sources of the emissions of harmful compounds can be accurately and reliably identified [16]. Low-molecular-weight PAH compounds are usually formed by low-temperature combustion processes, while high-temperature combustion is associated with the emission of higher molecular weight compounds [17]. High-temperature biomass combustion in energy processes causes the decomposition of organic compounds into reactive radicals, which form stable aromatic ring bonds during pyrosynthesis to form five- or six-ring PAH compounds [18]. Soil is a good indicator of environmental pollution and the risk it poses to humans [19]. Due to their persistence and hydrophobicity, *PAHs* remain in soil for a long time and therefore constitute the main reservoir of *PAHs* in the environment. Performing soil contamination studies is one way of monitoring the risk of human exposure to *PAHs*. Up to now, most of the research has concerned soils from agricultural land, with less attention having been paid to forest soils, especially those that have been intensively visited by people in the last several months of the pandemic.

The aim of our study was to evaluate the sources of, and ecological risks associated with, *PAHs* in soils under forested areas around urban agglomerations. The study covered a 150 km long transect in southern Poland, stretching from the Upper Silesian Industrial Region—the most polluted area—to Kraków, and included both urban and commercial forests that are intensively used for tourism and recreation. Although there have been many studies on *PAHs* in soils, most of these have focused on agricultural land. With economic development, the pollution characteristics have changed, and more systematic investigations and monitoring are needed, not only of agriculture soils, but also forest soils.

## 2. Materials and Methods

### 2.1. Study Area and Soil Sampling

The study was conducted in the Silesian and Lesser Poland voivodeships, located in southern Poland (Figure 1). The average annual precipitation in the area is 700–750 mm and the average air temperature is around 8.5 °C.

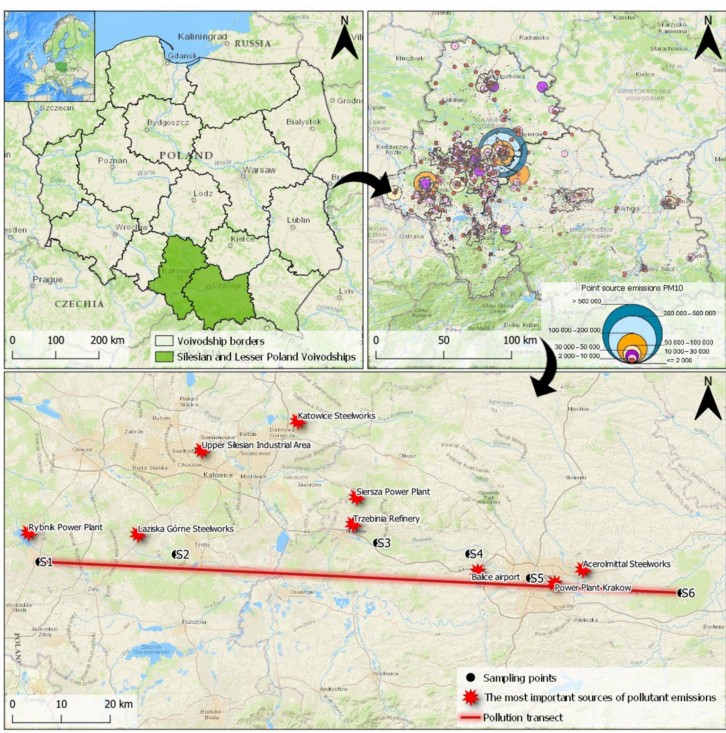

**Figure 1.** Location of the sampling sites.

This area is the most industrialised geographical region in Poland [20] and is associated with the occurrence and exploitation of hard coal deposits for the energy industry. Most of the mines are located in the Upper Silesian Industrial Region, where there are also several coal-fired power plants, industrial factories and coking plants. Due to the high industrial intensity associated with the energy economy, the study area has been strongly influenced by pollution. The study area is also characterized by the densest level of agglomeration and population, which undoubtedly has a huge impact on air quality and pollutant emissions [21,22]. According to the Environmental Impact Report prepared for the Silesian Voivodeship [23] and the Lesser Poland Voivodeship [24], emissions from Upper Silesian Industrial Region plants account for 53% of the national emissions of gaseous pollutants (excluding carbon dioxide) and about 25% of the national emissions of dust. In 2013–2018, the average annual concentrations of inhalable particulate matter (PM10) exceeded the permissible standard of 40 $\mu g \cdot m^3$ at stations in the Kraków Agglomeration and the City of Rybnik, among others. Exceedances of permissible values of daily concentrations of PM10 have occurred over the entire study area, most frequently in winter, and are mainly related to dust emissions originating from the heating of individual buildings. The annual average concentrations of benzo(a)pyrene (*BaP*) in PM10 at all stations in the Silesian and Lesser Poland Voivodeships have exceeded the target limit of 1 $ng \cdot m^3$.

Test plots were selected based on field observations. Sample plots with equivalent soil types (Cambisols) and soil textures (sandy loam) were then selected for analysis [25]. The sampling sites were located in stands of coniferous monocultures formed by Scots pine (*Pinus sylvestris*). The stands were all 80 years old. A 150 km long transect was chosen, containing six sampling sites, with their average distances being about 30 km from each other. The transect ran through areas with the highest concentrations of industrial pollutants in both commercial and urban forest zones (Figure 1). Six study plots were designated at each site. From all locations along the transect, a total of 25 soil samples were collected. In location S1, 3 replicates of soil samples were collected, while locations S2, S3, and S5 contained 6 replicates each. Locations S4 and S6 had 2 replicates of each sample. Samples were taken from the 0 to 10 cm horizon, after removal of the organic horizon.

## 2.2. Laboratory Analysis

For analysis of the PAH content, naturally moist, fresh samples were sieved (2 mm mesh) and stored in the dark at 4 °C prior to analysis. To determine the PAH compounds, 10 g of each soil sample were extracted using 70 mL of 2-propanol. These samples were then centrifuged (4500 rpm, 5 min) and the supernatants were collected and subjected to solid-phase extraction (5 mL·min$^{-1}$) using a CHROMABOND CN/SiOH column. The resulting residues were dissolved in acetonitrile and analysed by high-pressure liquid chromatography (HPLC) using a Dionex UltiMate 3000 HPLC system equipped with a fluorescence detector and a Dionex UltiMate 3000 Column Compartment C18 5 µm with a 4.6·100 mm HPLC column. The mobile phases were water (A) and acetonitrile (B) at a flow rate of 1 mL·min$^{-1}$. Calibration solutions of different concentrations (0.1, 0.2, 0.5, 1 and 2 µg·mL$^{-1}$) were prepared based on a PAH calibration-mixture standard (CRM 47940) of 10 µg·mL$^{-1}$. The prepared solutions were placed in a chromatographic column, with the obtained chromatograms used to prepare a calibration curve. The soil samples were then analysed in triplicate. After every ninth analysis, a control sample (calibration solution of 0.1 µg·mL$^{-1}$) was injected. The concentrations of naphthalene (Nft), acenaphthene (Ace), fluorene (Flu), phenanthrene (Phe), anthracene (Ant), fluoranthene (Flt), pyrene (Pyr), benzo(a)anthracene (BaA), chrysene (Chr), benzo(k)fluoranthene (BkF), benzo(b)fluoranthene (BbF), benzo(a)pyrene (BaP), dibenzo(ah)anthracene (DahA), indeno(1,2,3-c,d)pyrene (IcdP) and benzo(g,h,i)perylene (BghiP) were measured.

## 2.3. Toxicity Equivalent Quotient

Calculation of the toxicity equivalent quotients (*TEQs*) of the *PAHs* relative to *BaP* was used to determine an ecological risk assessment of the *PAHs*. This method is often used in aquatic environments, soils and sediments [26–28]. Seven toxic *PAHs* (*BkF*, *BaP*, *BbF*, *Chr*, *BaA*, *IcdP* and *DahA*) were considered for the calculations, based on the US Environmental Protection Agency (USEPA) protocol [29]. The formula used is as follows:

$$TEQ_{CARC} = C_{BaA} \cdot 0.1 + C_{Chr} \cdot 0.001 + C_{BbF} \cdot 0.1 + C_{BkF} \cdot 0.01 + C_{BaP} + C_{IcdP} \cdot 0.1 + C_{DahA} \quad (1)$$

The lowest risk concentration of *BaP* was converted to a *TEQ* of 0.0005 µg·kg$^{-1}$ (*TEQ$_{QV}$*) [30] to more clearly represent the risk level derived from the *TEQ$_{CARC}$* value. The risk classification is shown in Table 1.

**Table 1.** Risk levels for toxicity equivalent quotient (*TEQ*).

| Risk Level | *TEQ$_{CARC}$/TEQ$_{QV}$* |
|---|---|
| No risk | <0.1 |
| Low risk | 0.1–1 |
| Low-to-moderate risk | 1–10 |
| Moderate-to-high risk | 10–100 |
| High risk | ≥100 |

## 2.4. Assessment of the Toxicity of PAHs in Soils

*PAHs* can be absorbed by humans through the skin and respiratory tract and can cause skin and lung cancer, among other diseases. Long-term exposure to *PAHs* in the environment can cause a number of carcinogenic processes in humans. Toxic equivalency factors (*TEFs*) have been used to estimate the risk of exposure to individual and all *PAHs* to human health. *BaP* equivalent concentrations (*BaPeq*) were used to determine the toxicity at each site along the transect, and the TEFs for all 16 *PAHs* were selected and calculated according to the USEPA [29] and Nisbet and LaGoy guidelines [31]. The total *BaPeqs* were calculated using the following equation:

$$\Sigma BaPeq = \Sigma C_i \cdot TEF_i \quad (2)$$

where $C_i$ is a single PAH concentration and $TEF_i$ is the relevant *TEF*.

### 2.5. Potential Ecological Risk Assessment

Plants and aquatic environments are potentially at risk from PAH toxins that accumulate in the soil. A risk quotient (*RQ*) has been used to determine the risk of harmful PAH substances [32]. To calculate the ecological risk of the *PAHs* in the soil, qualitative values for the average of negligible concentrations (*NCs*) and maximum permissible concentrations (*MPCs*) were used. With these, the $RQ_{(NCs)}$ and $RQ_{(MPCs)}$ could be determined, as follows:

$$RQ_{(NCs)} = \frac{C_{PAHs}}{C_{QV(NCs)}} \tag{3}$$

$$RQ_{(MPCs)} = \frac{C_{PAHs}}{C_{QV(MPCs)}} \tag{4}$$

where $RQ_{(NCs)}$ is the lowest *RQ*, $RQ_{(MPCs)}$ is the highest *RQ*, $C_{(PAHs)}$ is the exposure concentration of an individual PAH, $C_{QV(NCs)}$ is the lowest average risk value for the PAH, $C_{QV(MPCs)}$ is the highest risk value for the PAH, and Σ*PAHs* is the sum of the *RQ* values for all 16 *PAHs*. Table 2 provides the minimum and maximum standard values for individual *PAHs*, whilst Table 3 gives the ecological risk criteria for the individual *PAHs* and Σ*PAHs*.

**Table 2.** Lowest and highest risk standard values for *PAHs*.

| PAHs | Lowest and Highest Standard Risk Values | |
|---|---|---|
| | NCs | MPCs |
| | (ng·kg$^{-1}$) | |
| Nft | 12 | 1200 |
| Ace | 0.7 | 70 |
| Flu | 0.7 | 70 |
| Phe | 3 | 300 |
| Ant | 0.7 | 70 |
| Flt | 3 | 300 |
| Pyr | 0.7 | 70 |
| BaA | 0.1 | 10 |
| Chr | 3.4 | 340 |
| BbF | 0.1 | 10 |
| BkF | 0.4 | 40 |
| BaP | 0.5 | 50 |
| DBahA | 0.5 | 50 |
| BghiP | 0.3 | 30 |
| IcdP | 0.4 | 40 |

**Table 3.** Risk quotient levels for individual *PAHs* and the sum of *PAHs*.

| Risk Level | Individual PAHs | | Risk Level | Total PAHs | |
|---|---|---|---|---|---|
| | RQ(NCs) | RQ(MPCs) | | RQ(NCs) | RQ(MPCs) |
| No risk | <1 | | No risk | =0 | |
| | | | Low risk | ≥1; <800 | =0 |
| Moderate risk | ≥1 | <1 | Moderate risk 1 | ≥800 | =0 |
| | | | Moderate risk 2 | <800 | ≥1 |
| High risk | | ≥1 | High risk | ≥800 | ≥1 |

### 2.6. Incremental Lifetime Cancer Risk Ratio

The main routes of human exposure to soil-related toxic PAH compounds are soil ingestion, inhalation and dermal contact. The incremental lifetime cancer risk (*ILCR*) ratio

provides an estimate of the potential cancer risk to humans [33,34]. The cancer risk for the three pathways was calculated according to the equations [35]:

$$ILCR_{ingestion} = \frac{C_{soil} \cdot IR \cdot ED \cdot EF}{BW \cdot AT \cdot 10^6} \cdot CSF \tag{5}$$

$$ILCR_{inhalation} = \frac{C_{soil} \cdot HR \cdot EF \cdot ED}{PEF \cdot BW \cdot AT} \cdot CSF \tag{6}$$

$$ILCR_{dermal} = \frac{C_{soil} \cdot SA \cdot AF \cdot ABS \cdot EF \cdot ED}{BW \cdot AT \cdot 10^6} \cdot CSF \tag{7}$$

where $ILCR_{ingestion}$, $ILCR_{inhalation}$ and $ILCR_{dermal}$ are the $ILCRs$ associated with the exposure pathways of ingestion, inhalation and dermal contact, respectively, $C_{soil}$ represents the $TEQ$ concentrations of the $PAHs$ in soil, $IR$ and $HR$ are the ingestion rate (mg·d$^{-1}$) and air inhalation rate (m$^3$·d$^{-1}$), respectively, $ED$ is the exposure duration (yr), $EF$ is the exposure frequency (d·yr$^{-1}$), $BW$ is the body weight, $AT$ is the average time (d), $PEF$ is the particle emission factor (m$^3$·kg$^{-1}$), $SA$ is the surface area of the skin (cm$^2$·d$^{-1}$), $AF$ is the relative skin adherence factor (mg·cm$^{-2}$), $ABS$ is the dermal absorption factor, and $CSF_{ingestion}$, $CSF_{inhalation}$ and $CSF_{dermal}$ are the cancer risk slope factors for $BaP$ via the different pathways (mg·kg$^{-1}$·d$^{-1}$). The total cancer risk is assumed to be the sum of the three different pathways [35–38]:

$$ILCR_{dermal} = ILCR_{ingestion} + ILCR_{inhalation} + ILCR_{dermal} \tag{8}$$

Table 4 provides the cancer risk parameters for evaluating dermal and ingestion exposure.

**Table 4.** Factors in the incremental lifetime cancer risk ratio (ILCR).

| Exposure Parameters | Units | Adults | Children |
|---|---|---|---|
| Ingestion Rate (IR) | mg·d$^{-1}$ | 100 | 200 |
| Inhalation Rate (HR) | m$^3$·d$^{-1}$ | 20 | 10 |
| Exposure frequency (EF) | d·yr$^{-1}$ | 350 | |
| Exposure duration (ED) | yr | 20 | 6 |
| Conversion factor (CF) | - | $10^{-6}$ | |
| Body weight (BW) | kg | 70 | 15 |
| Average time (AT) | d·yr$^{-1}$ | 25,550 | |
| Surface area (SA) | cm$^2$ | 5700 | 2800 |
| Skin adherence factor (AF) | mg·cm$^{-2}$ | 0.07 | 0.2 |
| Particulate emission factor (PEF) | m$^3$·kg$^{-1}$ | $1.36 \times 10^9$ | |
| Adsorption factor (ABS) | - | 0.13 | |
| CSF$_{inhalation}$ | mg·kg$^{-1}$·d$^{-1}$ | 3.85 | |
| CSF$_{ingestion}$ | mg·kg$^{-1}$·d$^{-1}$ | 7.3 | |
| CSF$_{demal}$ | mg·kg$^{-1}$·d$^{-1}$ | 25 | |

*2.7. Diagnostic Indicators for the Identification of Sources of PAH Pollution Emissions*

In this study, PAH diagnostic indicators determined the type of biomass combustion used in the energy-production process [Flt/(Flt + Pyr); BaA/(BaA + Chr); IcdP/(IcdP + BghiP)] [39–41] differentiated the origin of the pollutants resulting from transport traffic (BaP/BghiP) [42] and were used to distinguish the different sources of PAH emissions along the transect (Table 5).

**Table 5.** Indicators defining the sources of PAH emissions.

| PAH Ratios | Range | Source Type | References |
|---|---|---|---|
| Flt/(Flt + Pyr) | <0.4 | Petrogenic | [40] |
| | 0.4–0.5 | Fossil fuel combustion | |
| | >0.5 | Grass, wood, coal combustion | |
| BaA/(BaA + Chr) | 0.2–0.35 | Coal combustion | [39,41] |
| | >0.35 | Vehicular emissions | |
| | <0.2 | Petrogenic | |
| | >0.35 | Combustion | |
| IcdP/(IcdP + BghiP) | <0.2 | Petrogenic | [39] |
| | 0.2–0.5 | Petroleum combustion | |
| | >0.5 | Grass, wood and coal combustion | |
| BaP/BghiP | <0.6 | Non-traffic emissions | [42] |
| | >0.6 | Traffic emissions | |

*2.8. Statistical Analysis*

ANOVA was used to assess significant differences between the mean values of the properties. The relationship between the properties was established using Pearson's coefficient. Principal component analysis (PCA) was used to interpret factors in certain datasets. Agglomeration of the localisation into groups that differed in PAH content was performed based on Ward's method [43]. All statistical analyses were performed using R [44] and R Studio software (2020 version, PBC: Boston, MA, USA) [45]. Surfer 15 software (Version 15, LLC: Golden, CO, USA) [46] was used to express spatial variability and to generate maps of the studied indicators, and the kriging interpolation method was used to generate the maps.

**3. Results**

There was a differentiation of the PAH contents in the soils of the studied transect (Table 1). On average, the highest number of *PAHs* (average total = 52.75 μg·kg$^{-1}$) occurred in the soils from Site S1 (city forests in Rybnik). Significantly high PAH contents (mean sums = 47.13 and 44.67 μg·kg$^{-1}$, respectively) were also noted at Sites S2 and S3. Significantly lower PAH contents (mean sums = 12.49 and 12.41 μg·kg$^{-1}$, respectively) were recorded in the forest areas around Kraków at Sites S4 and S6. In the soils of Kraków's urban forests, the PAH contents did not differ significantly from the other sites (Table 6). Among the determined *PAHs*, the most numerous in the soils were Flu, Flt, Pyr and Chr, whilst the least numerous were Nft and Ace. The differences in the PAH contents depended on the number of benzene rings, as shown in Figure 2. Four- and five-ring *PAHs* were dominant in the studied soils. The highest number of four-ring *PAHs* was recorded in the soils of Kraków and its vicinity (Sites S4–S6), and the lowest in the soils of Site S3. The highest number of five-ring *PAHs* was recorded in the soils of Site S3, and the least in the soils of the forests in Rybnik and its vicinity (Sites S1 and S2) (Figure 2). The share of six-ring *PAHs* ranged from a few to a dozen or so percent, with the most recorded in the soils at Sites S2 and S3.

**Table 6.** Content of individual *PAHs* ($\mu$g·kg$^{-1}$) in the tested soils.

| Molecular Weight | Number of Rings | Name of PAH | S1 | S2 | S3 | S4 | S5 | S6 |
|---|---|---|---|---|---|---|---|---|
| LMW | 2-rings | Nft | - | 1.82 ± 0.00 [b] | 14.66 ± 6.33 [a] | - | - | - |
| | 3-rings | Ace | 2.60 ± 1.97 [a] | 3.81 ± 2.73 [a] | - | 0.82 ± 0.50 [b] | 3.12 ± 1.21 [a] | 0.87 ± 0.18 [b] |
| | | Flu | 79.56 ± 51.55 [ab] | 100.70 ± 142.99 [a] | 43.55 ± 38.15 [b] | - | - | - |
| | | Phe | 51.36 ± 40.39 [a] | 23.78 ± 19.88 [ab] | 15.13 ± 7.50 [bc] | 6.23 ± 1.58 [c] | 37.55 ± 3.03 [a] | 9.52 ± 2.77 [c] |
| | | Ant | - | 51.44 ± 44.35 | - | - | - | - |
| | 4-rings | Flt | 88.51 ± 68.40 [a] | 82.37 ± 56.05 [a] | 61.74 ± 53.06 [ab] | 24.04 ± 10.15 [b] | 61.17 ± 19.69 [ab] | 19.45 ± 7.44 [b] |
| | | Pyr | 84.53 ± 61.09 [a] | 64.89 ± 37.90 [ab] | 50.34 ± 12.54 [ab] | 21.91 ± 9.08 [b] | 58.61 ± 19.56 [ab] | 16.79 ± 9.50 [b] |
| | | BaA | 52.28 ± 37.59 [a] | 41.13 ± 30.40 [a] | 19.80 ± 8.71 [ab] | 10.23 ± 4.80 [b] | 26.13 ± 7.57 [ab] | 12.12 ± 3.49 [b] |
| | | Chr | 99.58 ± 78.96 [a] | 67.12 ± 17.13 [b] | 39.81 ± 12.16 [bc] | 19.26 ± 7.99 [c] | 54.75 ± 14.72 [b] | 23.48 ± 1.52 [c] |
| HMW | 5-rings | BbF | 66.80 ± 57.72 [a] | 46.67 ± 9.37 [b] | 54.52 ± 46.08 [ab] | 20.23 ± 10.69 [c] | 34.85 ± 8.70 [bc] | 20.99 ± 3.27 [c] |
| | | BkF | 26.44 ± 21.80 [a] | 16.38 ± 4.11 [ab] | 17.99 ± 12.90 [ab] | 7.20 ± 3.77 [b] | 4.57 ± 1.01 [b] | 7.15 ± 0.65 [b] |
| | | BaP | 35.74 ± 27.75 [b] | 24.03 ± 6.10 [bc] | 113.62 ± 193.92 [a] | 9.91 ± 5.77 [c] | 28.82 ± 10.84 [b] | 11.23 ± 0.62 [c] |
| | | DBahA | 8.71 ± 7.05 [b] | 16.12 ± 10.00 [a] | - | - | 0.53 ± 0.16 | - |
| | 6-rings | BghiP | 31.41 ± 33.94 [a] | 20.93 ± 6.14 [ab] | 43.83 ± 37.37 [a] | 8.53 ± 4.94 [b] | 9.95 ± 2.38 [b] | 7.06 ± 3.07 [b] |
| | | IcdP | 36.55 ± 33.36 [ab] | 75.77 ± 143.10 [a] | 44.78 ± 40.21 [ab] | 9.06 ± 5.26 [c] | 19.05 ± 5.36 [b] | 7.89 ± 2.90 [c] |
| $\Sigma$ | | | 52.76 [a] | 47.13 [a] | 44.67 [a] | 12.49 [b] | 28.26 [ab] | 12.41 [b] |

LMW—low molecular weight, HMW—high molecular weight; naphthalane [Nft], acenapthene [Ace], fluorene [Flu], phenanthrene [Phe], anthracene [Ant], fluoranthene [Flt], pyrene [Pyr], benzo(a)anthracene [BaA], chrysene [Chr], benzo(k)fluoranthene [BkF], benzo(b)fluoranthene [BbF], benzo(a)pyrene [BaP], dibenzo(ah)anthracene [DahA], bezo(g,h,i)perylene [BghiP], indeno(1,2,3-c,d)pyrene [IcdP]; small letters in the upper index of average values indicate significant differences between research points and the content of individual PAH compounds.

The ecological risks of the *PAHs*, expressed as the *TEQs*, differentiated the soils of the studied sites. The soils of Site S3 had a moderate-to-high risk, whilst the risk was low-to-moderate at other sites (Figure 3, Table 3). The BaPeqs also were distinct in the soils at the different sites, with the highest recorded at S3, whilst S1 and S2 were characterised by lower, but still high, BaPeqs (Figure 3).

The lowest *BaPeq* was recorded in the vicinity of Kraków in the soils at Sites S4 and S6. The *RQs*, used to determine the risk of harmful PAH substances, were strongly differentiated in the soils of the disparate sites (Figures 4 and 5), but with the highest values of $RQ_{(NCs)}$ and $RQ_{(MPCs)}$ indicating high risk in the soils at all sites. The highest $RQ_{(NCs)}$ and $RQ_{(MPCs)}$ values were recorded in the soils of Sites S1, S2 and S3 (Figures 4 and 5). However, a very low cancer potential, relative to the adult *ILCR* index, was determined for Site S4, where it was $6.21 \times 10^{-6}$. At Sites S1, S2, S5 and S6, the results were within the low-risk range, scoring $4.86 \times 10^{-5}$, $3.87 \times 10^{-5}$, $3.06 \times 10^{-5}$ and $1.30 \times 10^{-5}$, respectively (Figure 6).

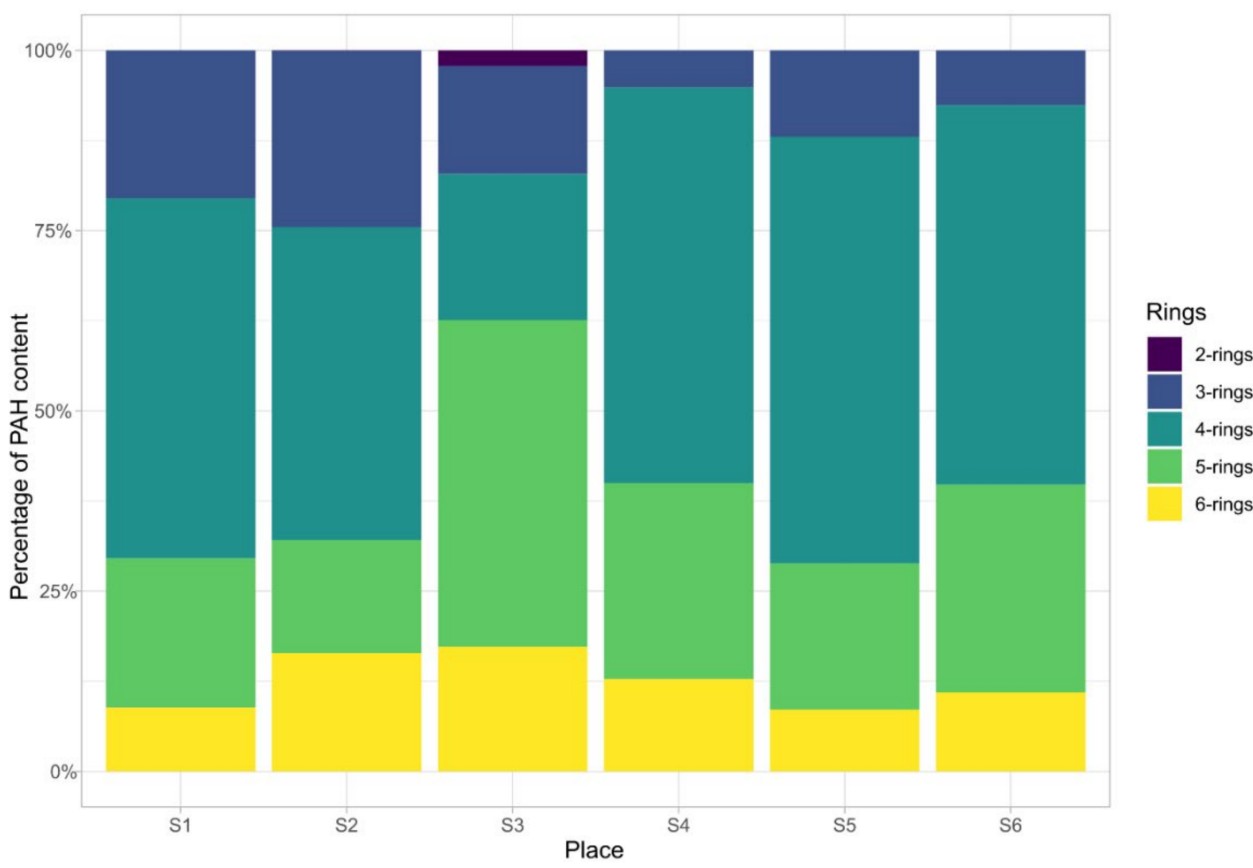

**Figure 2.** Mean percentage of individual *PAHs* in relation to the study plots.

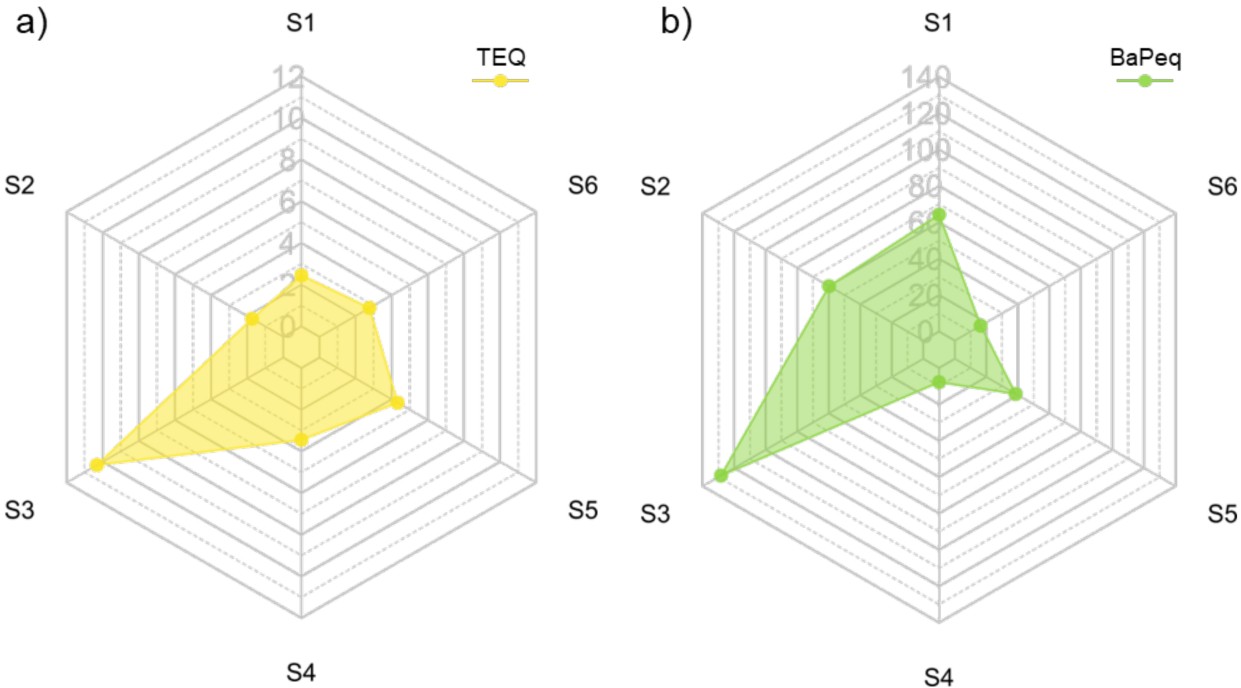

**Figure 3.** *TEQ* (**a**) and *BaPeq* (**b**) values of different sampling sites.

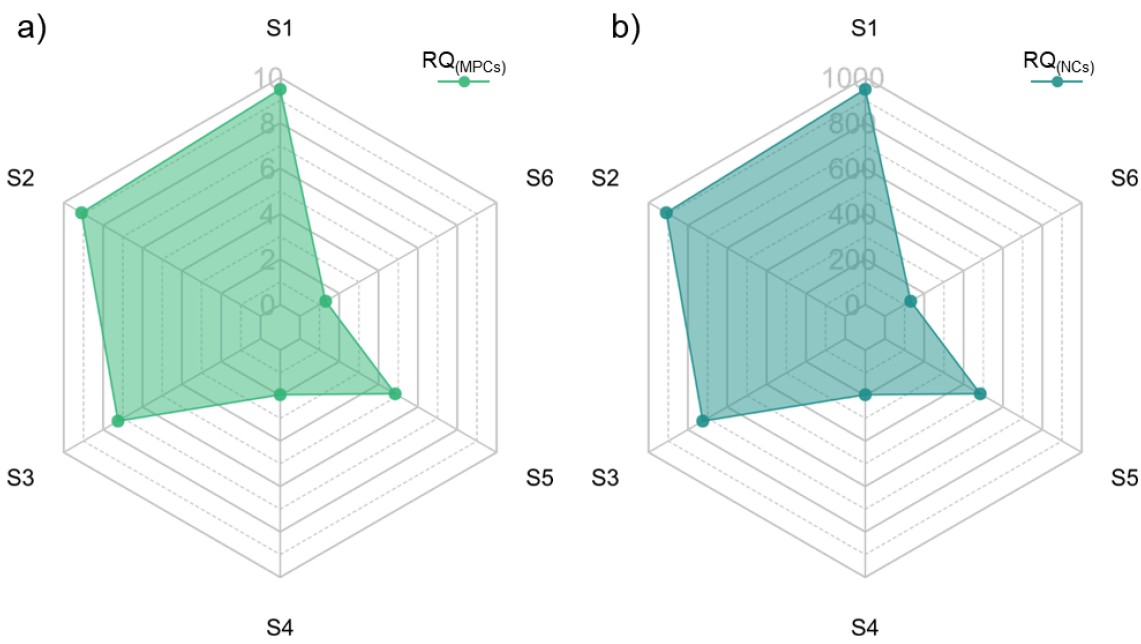

**Figure 4.** Lowest $RQ_{(MPCs)}$ (**a**) and highest risk $RQ_{(NCs)}$ (**b**) quotient values of different sampling sites.

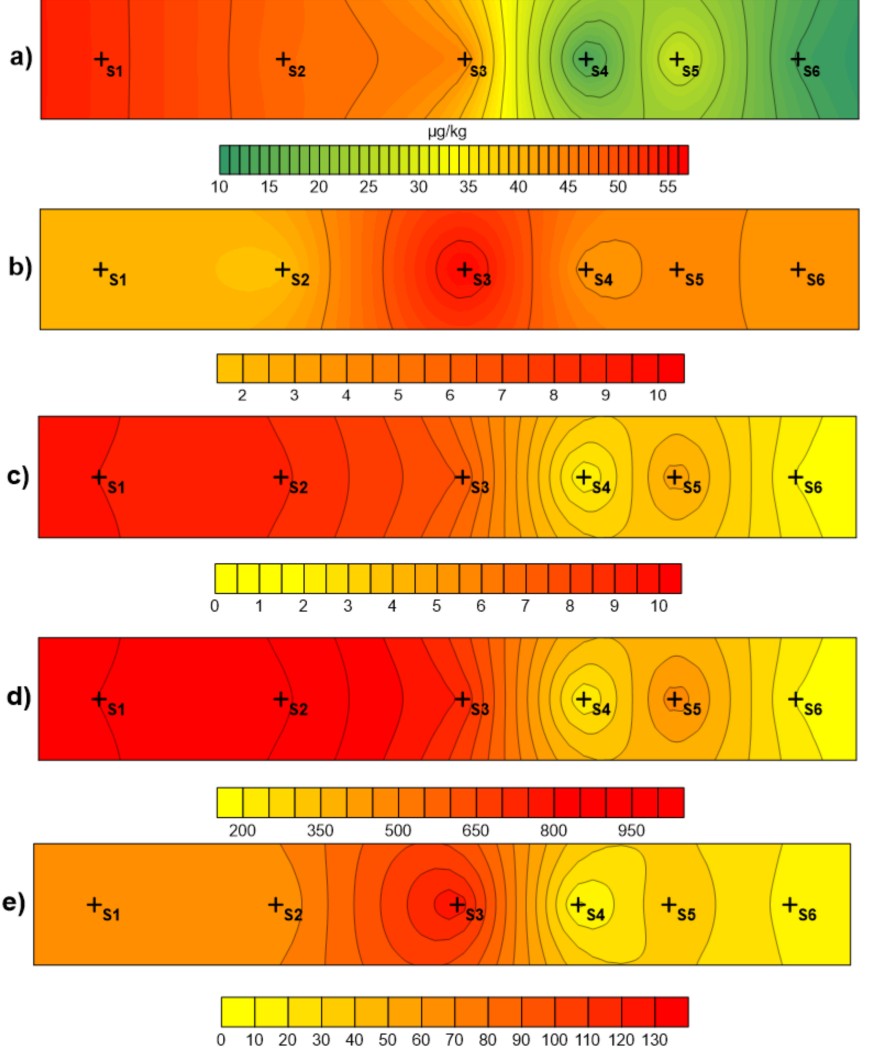

**Figure 5.** *Cont.*

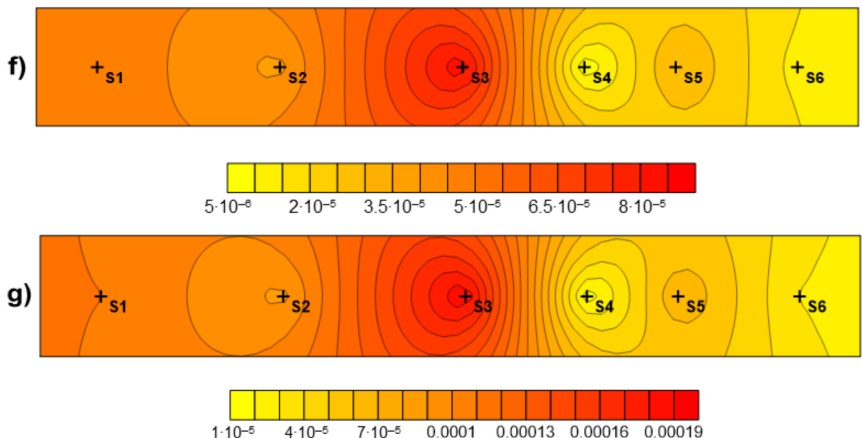

**Figure 5.** Maps of spatial distribution of (**a**) average PAH contamination; (**b**) *TEQ* value; (**c**) *RQ(NCs)* (**d**) *RQ(MPCs)*; (**e**) *BaP* equivalent; (**f**) *ILCR* adult; (**g**) *ILCR* children.

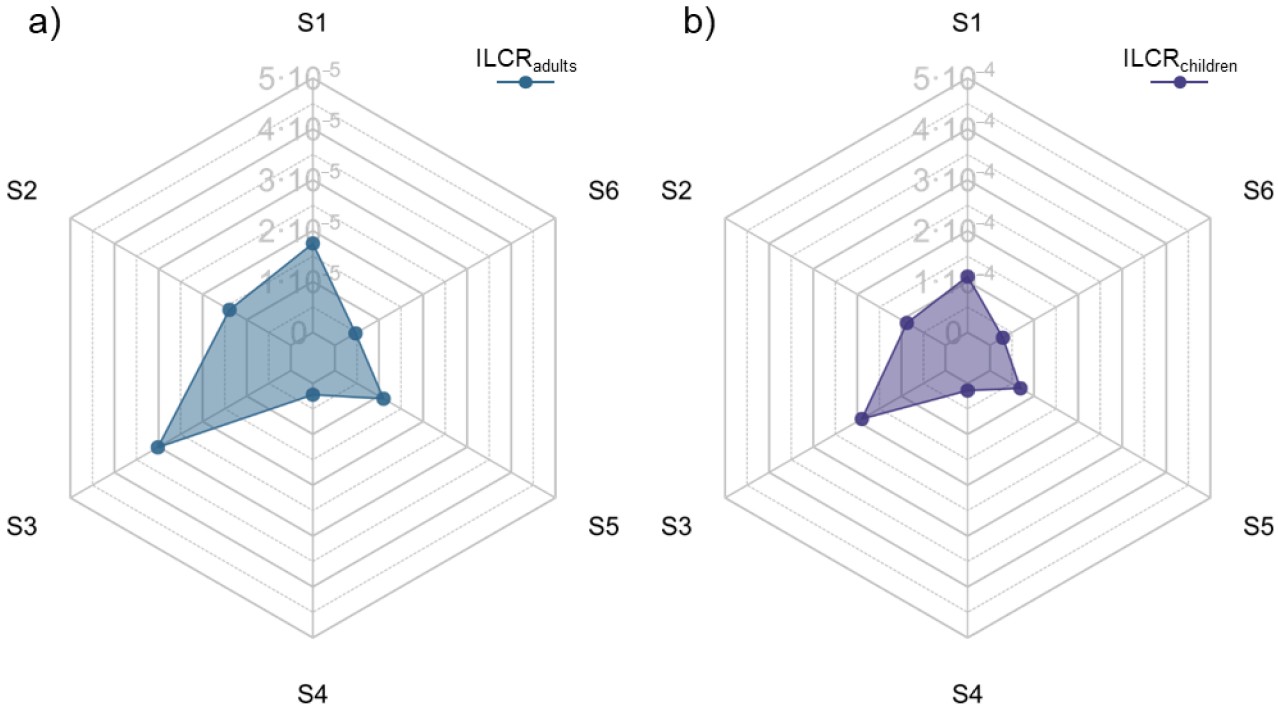

**Figure 6.** Incremental lifetime cancer risk ratios (*ILCR*) for adults (**a**) and children (**b**).

Site S3 had the highest recorded risk, at $8.38 \times 10^{-5}$, which places it closest to the moderate risk category of the *ILCR* index. By contrast, the *ILCR* index for children at Sites S1 and S3 reached the moderate risk category, with scores of $1.1 \times 10^{-4}$ and $1.9 \times 10^{-4}$, respectively. The remaining sites (S2, S4, S5 and S6) scored $8.76 \times 10^{-5}$, $1.41 \times 10^{-5}$, $6.94 \times 10^{-5}$ and $2.95 \times 10^{-5}$, respectively, thus falling in the low-risk range. In order to identify the sources of the *PAHs*, we used four diagnostic ratios. A comparison of the coefficients that determined the sources of the *PAHs* showed that the cause of soil contamination in all variants was mixed, with Figure 7 showing that the share of pollutants in the soils was affected by industrial combustion, biomass and coal combustion, and traffic emissions. The grouping analysis, performed using the *RQ, BaPeq, TEQ* and *ILCR* ratio values, confirmed the distinctiveness of the soils from Sites S1–S3 and S4–S6 (Figure 8).

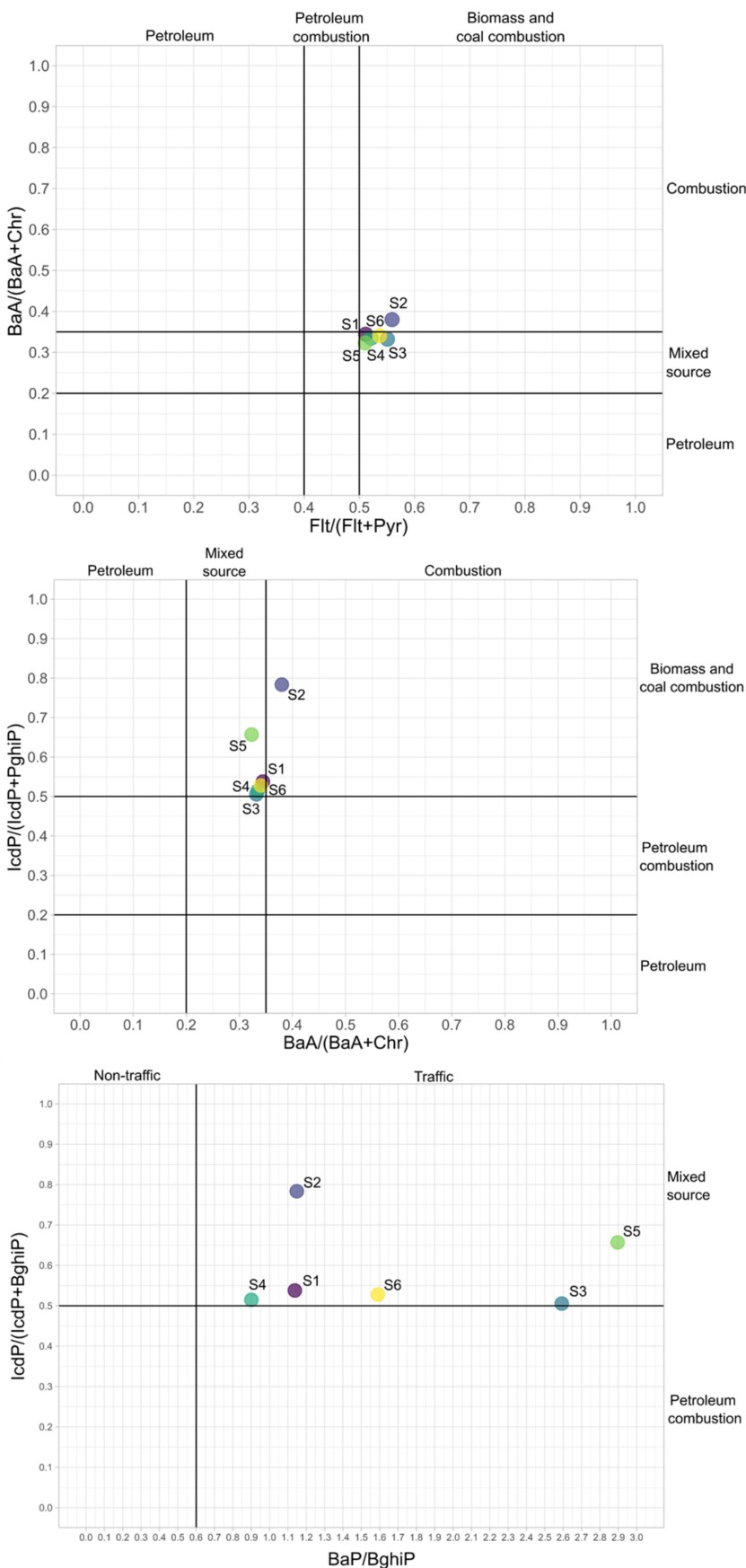

**Figure 7.** Diagnostic coefficients of PAH sources at different study sites.

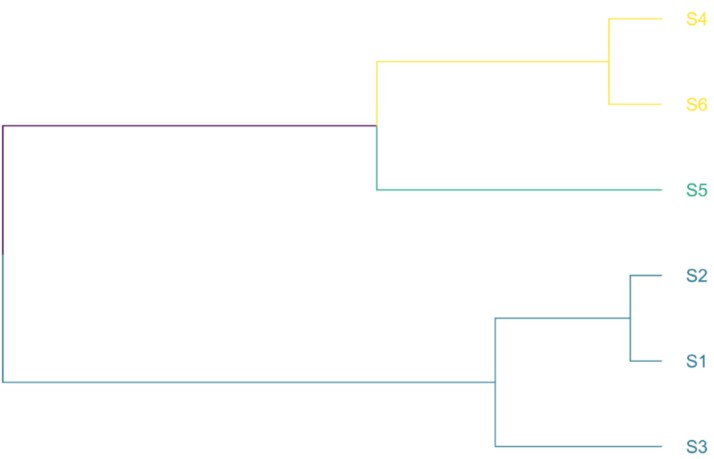

**Figure 8.** Dendrogram comparing the ecological risk potential (*RQ*), total toxicity equivalent concentrations (*BaPeq*), toxicity equivalent quotient (*TEQ*) and incremental lifetime cancer risk ratio (*ILCR*) of the studied soils.

Factors 1 and 2, distinguished by the PCA for the organic horizons, explained a total of 68.8% of the variance in the soil characteristics (Figure 9). The PCA also confirmed the distinctiveness of the soils, indicating that S3 was distinct from the others. Factor 1 was mainly related to the risk assessment indicators, while Factor 2 was related to the indicators that identified the source of pollution (Figure 9).

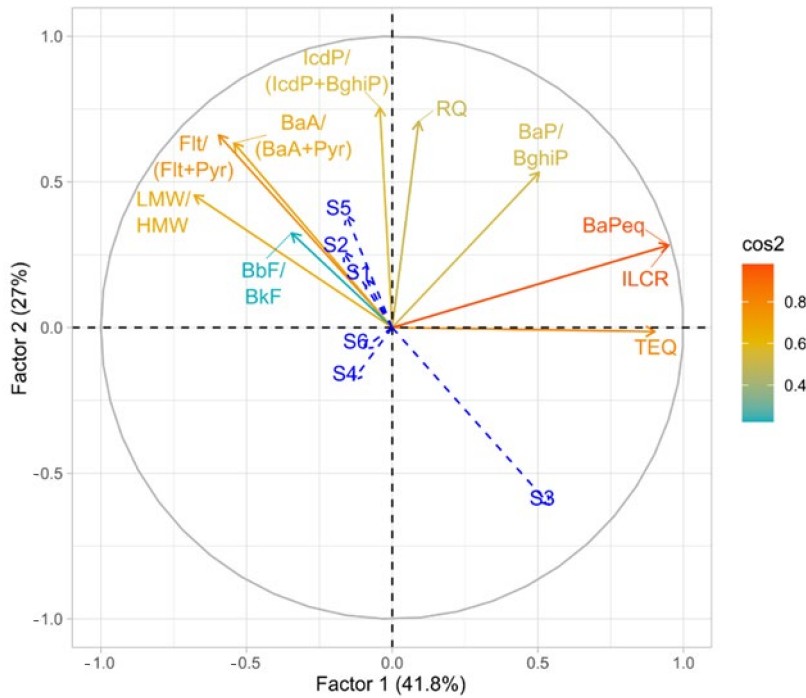

**Figure 9.** Projection of the variables on the factor plane.

## 4. Discussion

The PAH concentrations in the forest soil samples of the studied transect were compared with those from other Europe regions. This comparison revealed that the studied soils had relatively low PAH concentrations compared to the other regions. From this study, the mean sums of the *PAHs* ranged from 12.41 to 52.76 $\mu g \cdot kg^{-1}$, whilst the sum of 16 PAH concentrations from the topmost 2 cm of forest soils from a north–south transect through the city of Oslo had a maximum value of 2.6 $mg \cdot kg^{-1}$ dm [47]. According to Aichner et al. [48], the sum of 16 EPA-PAH concentrations from German forest soils ranged

from 105 to 14.889 ng·g$^{-1}$ dw in the O horizon [49], and from 20 to 9038 and from 7 to 4424 ng·g$^{-1}$ dw in the mineral topsoil (0–5 cm and 5–10 cm, respectively). The sum of 16 PAH concentrations in soils from London ranged from 4 to 66 mg·kg$^{-1}$, with a mean of 18 mg/kg and a median of 14 mg·kg$^{-1}$ [50]. The PAH distribution patterns showed that four- and five-ring PAH compounds dominated, whereas two- and three-ring PAH compounds were barely present at all. From the group of four-ring *PAHs*, the Flt, Pyr, BaA and Chr contents were determined. In this study, Flt, Pyr and Chr were present in the highest amounts. The BbF, BaP, IcdP and DahA contents were determined from the group of five- and six-ring *PAHs*. Hydrocarbons with fewer rings degrade faster than those with more, and *PAHs* are subject to various chemical reactions, including oxidation, reduction and electrophilic substitution. Although *PAHs* are relatively permanent compounds, they undergo reconstruction, for example, under the influence of thermal factors. In terms of environmental protection, oxidation reactions are important in the process of PAH degradation in the environment [51]. Hydrocarbons with a smaller number of rings in the molecule undergo faster microbial decomposition, which in turn leads to the undesirable accumulation of more harmful *PAHs* (with a greater number of rings) in the soil [52]. It should be understood that *PAHs* never occur individually in the environment, which means that a high concentration of one compound from this group indicates a high level for others as well, as was confirmed in our study.

The most contaminated soils from the studied transect were those from the Upper Silesian Industrial Region (Sites S1 and S2). This and the adjacent areas represent a critical example of the fabricated change of a natural system. The main cause of the catastrophic pollution of the Upper Silesian Industrial District is the excessive concentration of industry and mining, buildings and the corresponding development of communications and transport industries. The emission of *PAHs* into the environment from anthropogenic sources is several times higher here than the natural background emissions. It is believed that the combustion process is responsible for over 90% of the *PAHs* present in this environment [53], with the largest amount of *PAHs* entering the environment via the burning of fuels to heat homes and as a result of heavy-industry activities associated with the processing of coal and oil, mainly in the coke, petrochemical and metallurgy industries [54]. The forested area between Rybnik and Kraków had some of the highest 16-PAH concentrations. These high values can be explained by the predominant wind direction being towards the west, carrying *PAHs* from the Upper Silesian Industrial Region to the forest. However, more local sources, such as refineries and power plants, cannot be excluded. Our results show the emissions fingerprints of local and regional sources and suggest that these are major influencing factors in the composition of the *PAHs* in the study area. According to Wang et al. [19], the level of urbanisation corresponds very well to PAH accumulation in soils, although the main sources of the *PAHs* are different for different levels of urbanisation. The diagnostic indicators of PAH sources used in this study did not identify them unequivocally. In all cases, a mixed source of contamination was determined. Our analysis indicated that the share of pollutants in the soils was contributed to by industrial combustion, biomass and coal combustion, and traffic emissions. These findings may be related to the specificity of the studied transect, with the main cause of pollution in the region being industry and mining, communications and transport, and the fact that coal is still the dominant energy source in Poland [55].

The PAH risk assessment indicators employed herein were particularly useful for the forest soils, with the ecological risks of the *PAHs* found to be moderate-to-high. The highest risk was recorded in the soils at Sites S1–S3, in the Upper Silesian Industrial District or within its impact range. Earlier studies on the contamination of not only soil have indicated the utility of such risk assessment indicators [56,57]. Cancer risk is usually estimated using the *ILCR* index, which explains the increase in likelihood of developing cancer via pathways of exposure to potential carcinogenic compounds. The *ILCR* ratios based on the USEPA [28,35] and Li et al. [29] guidelines indicate that values below $10^{-6}$ represent very low risk, $10^{-6}$–$10^{-4}$ represent low risk, $10^{-4}$–$10^{-3}$ represent moderate risk and

$10^{-3}$–$10^{-1}$ represent high risk. According to Health Canada [58], the cancer risk is considered negligible if the estimated *ILCR* is 1 in 100,000 (i.e., $\leq 1 \times 10^{-5}$), but if the *ILCR* ratio is greater than $1 \times 10^{-5}$, risk management measures should be taken. Most of the sites in close proximity to the toxic emissions from the Silesian region fell within the $10^{-5}$–$10^{-4}$ range, indicating low-to-moderate lifetime cancer risk. Close proximity to the deposition of pollutants from the Upper Silesian Industrial District and the harmful PAH emissions from the Trzebinia Refinery contributed to the high *ILCR* values near Sites S1–S3. Several studies have also confirmed the extremely harmful impacts of refineries as major centres of PAH emissions into the environment, including their effects on human health [59,60] which is why management measures are recommended where the *ILCR* index is greater than $10^{-5}$ [58]. Exposure to toxic PAH compounds can initiate carcinogenic processes that can have a decisive impact on the subsequent course of COVID-19 disease, highlighting the comorbidity factor [61,62]. Our findings suggest the need for further monitoring of the quality of urban and suburban forest environments. Current forecasts indicate that, in 2050, 69.6% of the world population will live in cities. For many people, migrating to large cities is associated with the need to adapt to life in a new anthropogenic environment and includes finding places to spend free time that is as close as possible to the city.

## 5. Conclutions

Our findings confirm the potential in applying risk assessment indicators to forest soils. Here, they were able to differentiate between the soils from the studied sites. The assessment indicated that *PAHs* have strongly contaminated the soils of the Upper Silesian Industrial Region and the area under its influence. Less pollution was recorded in soils from the urban forests of Kraków and the forests around this agglomeration. Differences related to the degree of urbanisation were clearly distinguishable along the studied transect. However, the diagnostic indicators were not as useful for identifying the sources of the *PAHs* in the forest soils. Our analysis indicated areas where the PAH content may pose a risk to people's health, whilst the overall potential ecological risk of the soil *PAHs* was found to be high. As a result, the inhabitants of Rybnik and the surrounding area should consider spending their free time in forested areas somewhat removed from the Upper Silesian Industrial Region. It is clear that systematic and continuous PAH monitoring is necessary to keep people safe.

**Author Contributions:** Conceptualization, E.B. and J.L.; methodology, E.B., J.L. and S.Ł.; software, S.Ł.; validation, S.Ł. and E.B.; formal analysis, E.B., J.L. and S.Ł.; investigation, E.B. and S.Ł.; resources, E.B., J.L. and S.Ł.; data curation, E.B. and S.Ł.; writing—original draft preparation, E.B., J.L. and S.Ł.; writing—review and editing, E.B., J.L. and S.Ł.; visualization, S.Ł.; supervision, E.B. and S.Ł.; project administration, E.B. and S.Ł.; funding acquisition, E.B. and J.L. All authors have read and agreed to the published version of the manuscript.

**Funding:** Research for this paper was financed by a subvention from the Ministry of Science and Higher Education of the Republic of Poland for the University of Agriculture in Krakow for 2020 (SUB/040012/D019; A439).

**Data Availability Statement:** Data available on request from the authors.

**Conflicts of Interest:** The authors declare no conflicts of interest.

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
