# Peer review of "Ecological Risk Assessment, Distribution and Source of Polycyclic Aromatic Hydrocarbons in the Soil of Urban and Suburban Forest Areas of Southern Poland"

_forests, doi:10.3390/f15040595_

Round 1

Reviewer 1 Report

Comments and Suggestions for Authors

This article focused on the risk of PAHs that people were exposed to while spending leisure time in the forest during COVID-19. A 150km transect was selected to investigate, revealing the accumulation and environmental risks of PAHs in different regions. But there are still some questions to be resolved.

1.         What did the “all elements” mean in Line 43? Or “compartments”?

2.         What did the “the impact of sources on compartments” mean in Line 62?

3.         Sentence in Line 70-75 should be supplied with reference.

4.         The replicate number and total number of samples should be given.

5.         The map of sampling sites did not well reflect the relationship between the layout of sampling sites and PM10 emissions

6.         The influence of urban agglomeration on PAH was constantly mentioned in the introduction, and the role of urban agglomeration should also be highlighted in the following section.

7.         In line 214, Table 1 did not reflect the differentiation of PAH content.

8.         Whether there was significant changes in the mean percentage of individual PAHs, If they existed, it should be marked.

9.         In Figure 8, different diagnostic analyses indicated different sources of PAHs in sampling sites. How to ensure the objectivity of diagnosis.

10.     Although this paper made a detailed investigation on PAHs in the environment, I still think there were some things that need to be added. For example, a variety of indicators were used, but the results of PAHs sources revealed by different indicators were different, so it is necessary to extract a unified result from these indicators. In addition, although possible sources of PAHs were indicated in the diagnosis, there was a lack of information on the contribution of different sources to PAHs concentrations. In addition, the composition of PAHs at forest sites was the same as that at other sites. Did this indicate that the presence of PAHs at forest sites was caused by the input of wind or runoff?

11.     The conclusion showed that even the forest has the risk of PAHs accumulation. Can you propose some measures to reduce the risk of PAHs in the forest without sacrificing people's leisure time in the forest.

Author Response

Dear Reviewer,

Thank you very much for all the comments to improve the manuscript. Please accept the submission of a revised version of our manuscript. We would like to thank the Reviewer for your diligence and detailed revision, as well as your insightful and constructive comments. Please find our responses below, showing the changes we have made.

Attached to the system is a manuscript file with the changes annotated in yellow. Please find our responses below, showing the changes we have made.  

Looking forward to hearing from you, I remain respectfully yours,

PhD Stanisław Łyszczarz

Reviewer 2 Report

Comments and Suggestions for Authors

The article presented an important issue regarding soil pollution close to cities. In general, I think that the article gives a practical view of the situation in Central Europe.

The abstract is clear but, the keywords would be checked. One of the reasons is the size of the letter. However, as a suggestion, do not use the same words in the title and in the keywords to facilitate the search and dissemination of the paper.

In the Introduction, it seems that a problem with the size of the letter, like in the keywords, is presented. Considering the content of the introduction, I think that something about the persistence and stability of PAHs should be mentioned because this characteristic is one of the keys of their toxicity. This can enrich the number of citations included in the introduction too.

The section Materials and methods describes the sampling areas and the methods used properly. Moreover, although it is a not regular and straight transect, in my opinion it is quite good. I believe that accessibility is another important factor conditioning the selection of the sampling sites.

It is important no add the number of soil samples taken per plot. Six areas seem like a small number of areas sampled. It is desirable that there be a greater number of intermediate sampling points, but I imagine the difficulties in making them homogeneous both due to the type of soil and other conditions, such as vegetation.

I think that authors suppose that the data has a normal distribution to apply ANOVA (one way?). Regarding the Kriging used (in capital letter please as it is derived from a name), it is important to know which one was used, universal Kriging?

The results are clearly commented, although the same problem with the size of the letter was found in this section and the following. In general, there is a lot of information given.

I agree with the discussion. I think that if is a key point “to adapt to life in a new anthropogenic environment and includes finding places to spend free time that is as close as possible to the city”, municipalities have to do a great effort to adequate cities in a healthy way.

As minor comments:

Please try to avoid cutting the words at the end of a line in a way that makes it difficult to read, for instance, line 94 “south-ern”.

Tables surely, will be arranged later to avoid cutting them between pages,

Author Response

(The authors gave the same response as above.)
